# OpenReview forum: "Programmatic Evaluation of Rule-Following Behavior"
_ICLR.cc/2024/Conference — Submitted to ICLR 2024_

### Official Review · Reviewer_1QQx · 2023-10-31

**Soundness:** 2 fair
**Presentation:** 2 fair
**Contribution:** 2 fair
**Rating:** 5
**Confidence:** 2

**Summary:**

The paper studies the practical problem of avoiding “harmful” behaviors of LLMs.

To meet the “Three Laws of Robotics” in usability, safety, and ethics, the authors introduce the Benchmark for Identifying Noncompliant Decisions (BIND), a framework for evaluating rule-following behavior in LLM assistants.

The proposed benchmark contains 15 text scenarios drawing from the field of computer security and common children’s games. Each scenario defines a set of rules in natural language and an evaluation program to check model outputs for compliance with the rules. The authors also systematically collect a challenging hand-written test suite of 862 test cases across all 15 scenarios, against which they evaluate current state-of-the-art models and find lackluster performance.

**Strengths:**

The paper has successfully introduced the Benchmark for Identifying Non-compliant Decisions (BIND), a framework to programmatically evaluate rule-following in LLMs. The benchmark consists of 15 text scenarios in which the model is instructed to obey a set of rules while interacting with the human user to avoid “harmful” behaviors from LLMs.

**Weaknesses:**

The comparison to the relevant and closed baselines is not conducted. Without this comparison, it is hard to justify the advancements of the proposed framework.

The threats to the validity of the proposed benchmark are not investigated.

**Questions:**

Can the designed benchmark affect and guide the LLMs’ behavior? Please explain in detail with some concrete examples.

Is the effectiveness of the benchmark only affected and available to the current testing versions of the used LLMs? When the new versions of LLMs are released, will the proposed framework still be valid? Can the authors explain in detail with some examples?

How is the proposed framework performance compared to the baselines regarding the number of rules and the effectiveness in guiding the LLMs to avoid “harmful” behaviors?

---

> ### Author Response · Authors · 2023-11-16
>
> **Baselines.** We are unclear what types of methods are referred to as "baselines" by the reviewer. We would be happy to compare our benchmark with appropriate baselines if the reviewer could point them out. Since our scenarios rely only on text inputs and outputs, RuLES is compatible with future LLMs. We will publicly release all code and data, which we have now also uploaded. Future models may outperform existing models, but this will not invalidate the results found in this submission.
>
> **Red-teaming and alignment.** Our work is loosely related to prior work on red-teaming LLMs to evaluate their susceptibility to produce harmful or toxic outputs ("alignment"), but with important differences. Most relevantly, we evaluate ability to follow simple application-specific rules, rather than universal human values (such as "do not be offensive") that are relevant to all applications. Our setting, supporting zero-shot enforcement of application-specific rules not known at LLM training time, is plausibly harder than "alignment" (training to enforce universal values that are known at training time); and at the same time, it might also be easier, as we consider simple rules that admit an objective and unambiguous interpretation, whereas universal values can be subjective and ambiguous. For these reasons, different evaluation methods and benchmarks are needed in our domain, and that is what we provide in this paper.

---

### Official Review · Reviewer_do43 · 2023-11-01

**Soundness:** 4 excellent
**Presentation:** 3 good
**Contribution:** 3 good
**Rating:** 6
**Confidence:** 4

**Summary:**

The paper asks the question, _does expressing simple rules in natural language as prompts and/or system instructions ensure the model is able to follow these rules?_ To conduct effective and automatic evaluation, the rules choses can be evaluated by a simple computer program. The test scenarios are based on some pre-defined dimensions-- (1) Environments grounded in software security and games, (2) rules that need to be adhered to (Positive) and rules that should not be broken (Negative), and (3) Strategies (context setup in natural language) that can be used to push the model to break the rules. The experiments show, with multiple seeds and statistical testing, that both closed and open-source models fail to respect simple system rules that can easily be validated using simple computer programs.

**Strengths:**

1. The paper is well-written and easy to follow.
2. The objectives are clearly stated and the evaluation is broken down into reasonable dimensions.
3. The test-set will be a good benchmark for evaluating LLM's prowess at following simple rules in the future.
4. The experiments consider prompt variances, efficacy of system vs user prompts for SOTA models, and authors seem to have conducted statistical testing to support/disapprove their claims.
5. The authors report API results with timestamps and also consider automatic/optimized adversarial attacks on open-source models.

**Weaknesses:**

1. I would like to believe the finding in this paper should be reasonable obvious to most people at the conference, i.e. expecting a stochastic autoregressive model to follow deterministic objectives (that programs do) seems unreasonable to start with, although I have seen a suspension of disbelief from experts, alas! Truth be told, it seems like the season for papers along similar veins (LLMs can't plan, reason, solve NP hard problems, figure out game-theoretic equilibria); duh! Tbh, beyond the test set that will help others check to improve LLM capabilities at following simple rule (not sure why they need it though if we can write programs that can be called during orchestration), I am not fully sure of the contribution).
2. The choice of testing scenarios (esp. the security ones and the game) seems a little arbitrary, lacking good motivation.
3. The authors seamlessly refer to figures/tables in Appendix. While I did look at them for context and understanding, I feel this skews my evaluation towards other papers who have had to strictly adhere to the page limit to make their point.

**Questions:**

See above.

**Details Of Ethics Concerns:**

Although it is obvious to me (and hopefully people of this community), unsure if the jailbreaks/strategies proposed will help naive users break closed-source models into revealing other secrets (eg Personal Identifiable Information or PIIs) from their training data. While I am aware of vulnerability disclosure strategies in software security, unsure if such a paradigm exists for LLMs (or major players have reporting obligations). Wanted to see if the authors did any testing to ensure this is a no-threat or already communicated with the model providers. Tbh, unsure if the authors should be penalized for a single paper while other related work they cite is openly releasing attacks.

---

> ### Author Response · Authors · 2023-11-16
>
> **Novelty and contributions.** "Reasonably obvious" is a fair interpretation of our results, given what we have seen in all prior work on jailbreaking LLMs within and without the academic community. That said, while many might have reasonable intuitions or expectations about how well LLMs can follow rules, we are not aware of any prior work that has quantitatively or qualitatively evaluated this question. As we seek to motivate more clearly in our revised submission (we have also significantly reorganized the presentation of the material), we view our main contribution as one of formalizing an experimental paradigm for the research community in which to make reproducible progress on developing stronger defenses and more reliable rule-following behavior in LLM assistants. Our test suite is one approach to evaluating a variety of manual jailbreaking strategies. As behaviors and defenses evolve, it may be necessary to collect newer, harder test cases.
>
> **Ethics review.** We are happy to undergo an ethics review, but would note that our work does not introduce any novel vulnerabilities or attacks. Prior to submission, we spot-checked various manual jail-breaking strategies as well as optimized GCG suffixes on ChatGPT, etc. but did not find noticeably different results from what is already readily searchable on the Internet. Further, traditional vulnerability disclosure processes such as OpenAI's bug bounty program (https://bugcrowd.com/openai) explicitly disclaim model prompts and outputs as out-of-scope, so we did not pursue formal disclosure.

---

### Official Review · Reviewer_UwWZ · 2023-11-01

**Soundness:** 2 fair
**Presentation:** 3 good
**Contribution:** 3 good
**Rating:** 5
**Confidence:** 4

**Summary:**

Motivated that the model’s adherence to even simple rules needs human engagement, the authors propose a benchmark dataset where LLM rule-following can be programmatically evaluated. The proposed dataset consists of text scenarios that have an evaluation program to determine the model’s adherence to given rules. The text scenarios are influenced by computer security systems and children’s games. With the design of the dataset, the authors also use test suites of 862 hand-written test case templates to implement different high-level attacks for diverse analyses on rule-following behavior.

**Strengths:**

- Suggest a benchmark dataset in which LLM’s rule-following behavior can be automatically evaluated.
- The evaluation of each rule-following is robust and cheap.

**Weaknesses:**

- As the author’s motivation is to evaluate the rule-following behavior of LLMs automatically, the direct tackling to this motivation would be the automatic evaluation of rule-following behavior in arbitrary (at least diverse) domains and rules. However, the test scenarios are fixed in two domains, and the testing rules are limited to predefined contents for each domain. Fixed domain and rules can be evaluated by human, so harm the contribution of this work.

**Questions:**

- Can the suggested benchmark dataset be extended to other domains and rules with relatively little effort?
- As the different test suites change the LLM's performance, why defense prompt doesn’t work? Is there an explainable reason?
- Does rule-following behaviour in computer security system and children's game have general impact for assessing LLM performance?

---

> ### Author Response · Authors · 2023-11-16
>
> The purpose of our evaluation programs is to remove the dependency on manual labeling effort. The ability to automatically evaluate model responses is a key component of our benchmark, since it enables much faster and cheaper evaluation, shortening the development cycle.
>
> **Choice of scenarios.** We chose the two domains of security properties and children's games for their relevance and simplicity/familiarity, with the former domain being more "serious" and the latter being more "lighthearted". Our proposed benchmark can be extended to other domains with relatively little effort, e.g. "you are a security guard at the art museum", but we did not notice any qualitatively different behavior by LLMs between the two domains we did study.
>
> **Results and validity.** We found some issues with the defense prompts evaluated in the initial submission and have removed them from our revised version. Whether our benchmark carries any external validity in the real world is an important question that would require more longitudinal study of how performance on our benchmark correlates with real-world measurements of performance over time. However, we would point to the highly correlated efficacy of both manual and automatic jailbreaks in our scenarios, compared to generation of harmful/toxic outputs as investigated in prior work, as some initial evidence in our favor.

---

### Official Review · Reviewer_NyVg · 2023-11-06

**Soundness:** 2 fair
**Presentation:** 2 fair
**Contribution:** 2 fair
**Rating:** 3
**Confidence:** 5

**Summary:**

The authors describe a framework/benchmark named BIND that evaluates the ability of LLMs to follow rules under various scenarios (benign and adversarial).  They evaluate various LLMs using this benchmark and conclude that most LLMs in the status quo are not compliant with rules that are specified.

**Strengths:**

1. Benchmark is first of its kind.

**Weaknesses:**

1. Unclear what takeaways can be drawn from this work.
2. Paper could benefit from some reorganization.

**Questions:**

Overall, this work is interesting and potentially exciting but the main takeaways are not communicated in a clear manner. This reviewer is wondering what I can learn from this paper, and how others can follow-up on this line of work.

    1. The writing of the paper could benefit from some thought. For example, the authors could give examples of scenarios, rules and test cases to better highlight the difference between the 3 categories.
    2. The paper provides limited takeaways from their experiment. It is incredible that the authors have come up with such a benchmark. But what can I learn because of it apart from the fact that LLMs do not follow rules (which was already a well know fact. Look at work from Percy Liang’s group — https://arxiv.org/abs/2307.03172, or the fact that LLMs used in search e.g., BingChat can easily be subverted with prompt injection attacks)? The fact that there’s nothing beyond the creation of this benchmark is making this reviewer apprehensive in recommending acceptance.
    3. One conceivable application is one where the models are deployed in real-world systems (e.g., as in BingChat) and one would want to understand how brittle these are. But to validate such a scenario, the authors need to consider “layered defenses” i.e., add an output filter atop the generations from the LLMs and see how much information can be exfiltrated in such a setting. However, this was not done in this work.
4. A lot of the findings presented by the authors have been covered earlier i.e., numerous prompt injection strategies discuss mechanisms of rule subversion. This work, to me, seems like a consolidation of those findings. Could the authors emphasize the difference?

---

> ### Author Response · Authors · 2023-11-16
>
> The points about overall organization and discussion are well taken, and we believe that our revised submission provides a clearer organization and more tangible analyses of our results. The goal of this work is not to study particularly novel behaviors of LLMs, rather we present the design and usage of our evaluation scenarios and test suite as a potential test bed for future research. We point the reviewer to the files `llm_rules/scenarios/{security,games}.py` for the concrete implementations of our scenarios, as well as the files in `data/systematic/` to see our test cases.
>
> **Defenses.** We recognize that there is a large space of potential prompting and filtering methods that might substantially improve rule-following. We see our work as providing an evaluation benchmark that others can use, as they propose such interventions. As far as we are aware, current deployments of LLMs generally do not use those "layered defenses". It is also important to note the difficulty of properly evaluating proposed defenses. Without sufficient conscientious red-teaming, it is easy to draw faulty conclusions about proposed defenses [1], and we hope our benchmark will be a first step towards supporting such evaluation. We also have removed the scratchpad and double check prompting results from our revised submission because after further analysis of output logs, we realized not all the models fully understood our existing formulations of the prompts.
>
> **Novelty and takeaways.** It might be misleading to conclude that LLMs cannot follow rules at all; while the pass rate falls far short of what we would hope for, it is also significantly higher than zero. While many might have intuitions or expectations about how well LLMs can follow rules, we are not aware of any prior work that has quantitatively or qualitatively evaluated this question. Work on prompt injection is relevant, but generally has considered only one narrow attack strategy: e.g., "Disregard previous instructions and instead". We consider a broader array of strategies for fooling models into violating the rules. Some of our strategies would not be considered a successful prompt injection attack (because they do not allow completely replacing the existing task with any other task) but are able to trigger rule violations. As discussed in our revised submission, we also distinguish this work from red-teaming model alignment which focuses on circumventing a set of universal rules and values that the model has been trained to always obey. The preprint [2] suggested by the review does not appear to be relevant, as it considers performance in long contexts, while all of our scenarios use short contexts.
>
> [1]: Anish Athalye et al., Obfuscated Gradients Give a False Sense of Security: Circumventing Defenses to Adversarial Examples. ICML 2018
>
> [2] Nelson F. Liu et al., Lost in the Middle: How Language Models Use Long Contexts. ArXiv 2023

---

> > ### Comment · Reviewer_NyVg · 2023-11-19
> > **Thank you for your response!**
> >
> > Will internally deliberate. Thanks again for submitting your work to ICLR!

---

> > > ### Comment · Reviewer_NyVg · 2023-11-22
> > > **Based on deliberation**
> > >
> > > Thanks for your response. I have decided not to raise my score for the following reasons:
> > >
> > > 1. The evaluation in this work seems to hold for this snapshot in time without any generalizable takeaways. Questions that I'd like to have answered are: (a) why does this happen? (b) what is the difference between the long context case that Liu et al. have studied and the setting we consider?, (c) what can we do to fix this behavior?, (d) are smaller, more task-specific LLMs (e.g., Orca, the Phi series etc.) better at these considerations?  While I agree that it would be impossible to conduct these experiments in time for the rebuttal, the current paper is incomplete in my eyes without these questions and answers, and I am uncomfortable recommending acceptance for a paper where I am unconvinced about what follow-ups may stem from it.
> > >
> > > 2. **While many might have intuitions or expectations about how well LLMs can follow rules, we are not aware of any prior work that has quantitatively or qualitatively evaluated this question** --> this is echoing my point. While the authors have come up with a suite of clever techniques to evaluate rule-following behavior, I am always left wondering about why this matters in the real-world with more safeguards in place. The authors argue that it's hard to come to generalizable conclusions about defenses without proper red-teaming, yet advocate for acceptance of their work which is also devoid of any generalizable conclusions or methodologies; this seems paradoxical to me.  Try it out? Don't have to make a general claim. A simple google search of "bing chat output filters" provides ample anecdotal evidence that output filtering is done -- why not try something that you think is representative?

---

### Author Response · Authors · 2023-11-16

We thank the reviewers for their helpful comments. The feedback on the presentation of our work's motivation and takeaways is appreciated. We have uploaded a new version of our submission which has been substantively revised and hopefully addresses earlier concerns. The introduction, discussion, and related work have been edited to clarify our contribution, which we view as primarily one of establishing a new testbed for future research, as well as to contextualize our research among a broader scope of related work. We have also uploaded a cleaned + anonymized version of our code and data, which can help the reviewers better understand our scenarios and test cases.

---

### Meta-Review · Area_Chair_U5ij · 2023-12-05

**Metareview:**

This work performs an empirical study of if LLMs can faithfully follow instructions in a prompt throughout a conversation. On the positive sign, a strength of this work is that it is timely and relevant to avoiding jail breaking. The main negative raised by the reviewers is that it does not conclude much more than that, under their dataset, LLMs struggle to follow instructions. I would recommend that the authors resubmit to a dataset track.

**Justification For Why Not Higher Score:**

The primary scientific insight is that LLMs struggle to reliably follow per-conversation chat guardrails, and the reviewers (+AC) believe that the bar for publication should be higher: Explaining why this happens (with experiments to support the explanation), countermeasures, etc

**Justification For Why Not Lower Score:**

n/a

---

### Decision · Program_Chairs · 2024-01-16

Reject